# Loneliness and Satisfaction with Life among Nursing Students in Poland, Spain and Slovakia during the COVID-19 Pandemic

**DOI:** 10.3390/ijerph19052929

**Published:** 2022-03-02

**Authors:** Ewa Kupcewicz, Marzena Mikla, Helena Kadučáková, Elżbieta Grochans

**Affiliations:** 1Department of Nursing, Collegium Medicum, University of Warmia and Mazury, 11-041 Olsztyn, Poland; 2Department of Nursing, University of Murcia, Campus de Espinardo, 30100 Murcia, Spain; marmikla@yahoo.com; 3Department of Nursing, Faculty of Health, Catholic University in Ruzomberok, 034-01 Ruzomberok, Slovakia; helena.kaducakova@ku.sk; 4Department of Nursing, Pomeranian Medical University in Szczecin, 71-210 Szczecin, Poland; grochans@pum.edu.pl

**Keywords:** loneliness, satisfaction with life, student, pandemic

## Abstract

(1) The COVID-19 pandemic poses threats to human life and health around the world. This study attempts to determine the correlation between loneliness and satisfaction with life among nursing students in Poland, Spain and Slovakia and to seek predictors of social and emotional loneliness among the students. (2) A total of 756 nursing students from Poland, Spain and Slovakia took part in the study. A diagnostic survey was applied as the research method, and the data were collected with the sense of loneliness measurement scale (de Jong Gierveld Loneliness Scale–DJGLS) and the Satisfaction with Life Scale (SWLS). (3) The mean age of the participants was 21.20 years (SD = 1.97). A correlation analysis revealed statistically significant, negative correlations, with an average and high strength, between the general sense of loneliness and its components (social loneliness and emotional loneliness) and satisfaction with life among students in Poland, Spain and Slovakia. A regression analysis showed one’s satisfaction with life to be a predictor of a sense of loneliness among nursing students in Poland, Spain and Slovakia in the second year of the COVID-19 pandemic. (4) Students with a stronger sense of loneliness also feel lower life satisfaction. It is important to take preventive and prophylactic actions concerning loneliness among students during and after the COVID-19 pandemic.

## 1. Introduction

Due to the complexity of the state of loneliness as well as a multitude of interdisciplinary approaches, the concept is interpreted in a variety of ways. According to De Jong Gierveld et al., loneliness is a subjective and negative feeling. It is a consequence of a cognitive evaluation of the number and quality of existing relations and the individual’s attitude towards those relations. Loneliness can be one of the possible consequences of the situation evaluation, which manifests itself in a small number of relations. Whether an individual feels lonely because of the small number of contacts depends on the person’s standards [1].

Dołęga claims that young people’s loneliness is not a uniform psychological variable and can reveal itself in various areas of subjective feelings. It usually occurs in the emotional, social and existential spheres [2]. Emotional (subjective) loneliness is interpreted as a deficit of positive feelings and relations with people of special importance to an individual and, also, it is a feeling of a lack of close, satisfactory emotional bonds [3]. Dołęga defines a sense of social loneliness as the absence of social bonds, of belonging to a community, experiencing isolation and marginalization [3]. According to the author, existential loneliness denotes a lack of identification with generally accepted values, life goals and standards [3].

The COVID-19 pandemic has made people all over the world change the way they function in a very short time. There is growing concern that the restrictions and orders to “stay at home” because of the COVID-19 have led to an increasing sense of loneliness, which is regarded as a serious public health issue. Studies have shown links between loneliness and increased risk of mental illnesses, including depression, generalised anxiety and suicidal ideations [4]. Multiple studies have shown that a sense of loneliness is recently a consequence of restrictions imposed on society in connection with the SARS-CoV-2 virus spread [5,6]. It has been estimated that at least 38% to 50% of young people aged 18–24 years feel a higher level of loneliness during the obligatory lockdown than older ones [4,5]. Other scholars have observed a higher loneliness spread rate among young people during the COVID-19 pandemic compared to older people [7]. Communities of nursing students studying in the full-time system are most frequently composed of young people below 30 years of age. According to Spanish data, women and young adults are more susceptible than other groups to loneliness during the COVID-19 pandemic [8]. Polish studies have shown that two out of three women have experienced loneliness during the pandemic [9]. The findings of other studies have shown a strong sense of social loneliness in 22.7% of students under study, and 16.7% reported a strong sense of emotional loneliness [10]. Many studies have focused on seeking predictors of emotional and social loneliness. A study conducted by Labrague et al. among students in the Philippines [11] is an example. The research found that loneliness dominated among the students during the pandemic, and resistance, social support and coping behaviours were identified as factors protecting students from loneliness [11]. Although it is difficult to compare the presented results of studies from different countries due to different research methods applied and different understanding of the concept of loneliness, it can be concluded from the studies conducted in the first and second year of the COVID-19 pandemic that loneliness may become one of the most serious long-term problems related to the pandemic. Moreover, when predicting the effects of the pandemic, it should be assumed that loneliness impacts the mental, physical and social health of the entire population [12,13,14,15,16,17].

The period of the pandemic is an atypical, unusual and difficult period in many regards because the COVID-19 pandemic and the restrictions associated with it have changed everyday life, which seemed to be fixed and permanent. The COVID-19 pandemic presents new challenges for the theoretical and clinical education of nursing students. The study findings show that the SARS-CoV-2 virus spread has resulted in a considerable deterioration of students’ well-being [18,19]. It can be assumed that a sudden change from full-time nursing education at a university to distance learning resulted in a subjective feeling of dissatisfaction among the students, among others, in the area of their own social relations. Young adults use different coping strategies to offset the effects of isolation [20].

Satisfaction with life, manifesting itself in a subjective experiencing of positive and/or negative feelings, is an element of one’s good well-being [21]. It is associated with pursuing important goals, whose attainment, or even the path to them, can be a source of satisfaction [22,23]. There are many literature reports on the evaluation of satisfaction with life, which results from a comparison of one’s own situation with one’s own standards [24,25,26]. It can be presumed that life satisfaction is one of the main predictors of loneliness, considering the variety of applied restrictions related to the COVID-19 pandemic and cultural conditions in different countries, as well as the assumptions of loneliness and life satisfaction theories [27]. There is a lack of international research showing how and to what extent life satisfaction translates into the social and emotional functioning of nursing students during a pandemic. In contrast, some variables (e.g., related to social isolation) have been previously studied in the context of pandemics and have been reported in the global literature [28,29,30].

This is confirmed by the findings of Eweid et al., which showed that the most stressful factors for nursing students during the clinical classes during the COVID-19 pandemic were associated with a risk of infection and virus transmission to a relative. The majority of the students also reported tension and depression since the pandemic began [18]. According to Taylor et al., nursing students can face obstacles associated with the choice between performing their professional and ethical duty and the risk of infection [19]. Karabağ Aydın and Fidan showed in a Turkish study that a fear of death had a negative impact on satisfaction with life during the COVID-19 pandemic. A higher level of fear of death among nurses was associated with a lower level of satisfaction with life [26]. It was confirmed in another study that anxiety, mental health and social isolation were the main, statistically significant factors with an indirect impact on the quality of life of Turkish nurses [31].

Referring to the theory, which is the foundation for the construct of loneliness and its links with other variables, one can expect a link between a sense of loneliness and satisfaction with life in a group under study.

This study attempts to determine a correlation between loneliness and satisfaction with life among nursing students in Poland, Spain and Slovakia and to seek predictors of social and emotional loneliness among the students as a consequence of the spread of the SARS-CoV-2 virus.

In view of the presented reports, it was hypothesised that during the COVID-19 pandemic, more strongly experienced loneliness (understood both in terms of social loneliness and emotional loneliness) by nursing students was manifested by lower satisfaction with various aspects of life, compared to students who manifested higher levels of life satisfaction.

## 2. Materials and Methods

### 2.1. Settings and Design

The study was conducted by the diagnostic survey method, with the questionnaire technique, between 20 March and 15 May 2021. A total of 756 nursing students in the first degree (bachelor) studies who had hybrid classes at the University of Warmia and Mazury in Olsztyn, the Pomeranian Medical University in Szczecin (Poland), as well as at Murcia University (Spain) and the Catholic University in Ružomberok (Slovakia) were invited to participate in the study. The students’ age under 30 years was the enrolment criterion, and those who refused to grant their informed and voluntary consent for participation in the study were excluded. The survey in Poland and Spain was conducted at universities where teaching took place with a sanitary regime observed due to the spread of the SARS-CoV-2 infection. Hard copies of the questionnaire sets were distributed by one of the researchers. Due to the sanitary restrictions and the fact that nursing classes were conducted online, questionnaire sets were sent to the students at the Slovak university by e-mail, and when completed, they were returned by the same route within two days. After the deans’ consent was obtained in all the three universities, students were informed about the objectives and scope of the study and how the questionnaires should be completed. Students could ask questions, receive comprehensive answers and could withdraw from the study at any time, without providing a reason and with no consequences. Participation in the study was voluntary and anonymous. It took approximately ten minutes to complete the questionnaire.

Altogether 850 questionnaire forms were distributed, and 756 (i.e., 88.94%) correctly completed questionnaires were taken for the statistical analysis. The accumulated empirical data were encoded in an Excel spreadsheet, and a cumulative analysis was performed. This study is part of an international research project executed within the framework of one of the researcher’s (E.K.) scientific internship programme [28,32]. This project was given a favourable opinion (No. 3/2021) by the Senate Scientific Research Ethics Committee at the Olsztyn University in Olsztyn and was carried out in accordance with the Helsinki Declaration and the procedures and instructions currently in force at the university.

### 2.2. Participants

A total of 756 nursing students participated in the study, including 390 in Poland (51.59%), 196 in Spain (25.93%) and 170 in Slovakia (22.49%) [28,32]. The mean participant age was 21.20 years (SD = 1.97). Women accounted for a large majority of the participants (*n* = 682; 90.21%). Almost all participants described their health status as very good or good. Detailed characteristics of the participants are shown in Table 1.

### 2.3. Research Instruments

The study was conducted by the diagnostic survey method, and two standardised research tools were used to collect the empirical data. The tools are validated and available for use in the native language of each of the countries:A scale for measuring a sense of loneliness (De Jong Gierveld Loneliness Scale–DJGLS) developed by J. de Jong-Gierveld and F. Kamphuis [33,34];Satisfaction With Life Scale (SWLS) Ed Diener, Robert A. Emmons, Randy J. Larsen, Sharon Griffin [24].

An original survey questionnaire was used in the study to describe the group’s characteristics, which included the following socio-demographic data: place of residence, gender, age, study year, as well as information on selected elements of a student’s lifestyle (i.e., a restriction of physical activity, meals, time spent in front of a computer) during the COVID-19 pandemic.

#### 2.3.1. De Jong Gierveld Loneliness Scale (DJGLS)

The De Jong Gierveld Loneliness Scale developed by J. de Jong-Gierveld and F. Kamphuis was used to evaluate the sense of loneliness. The scale comprises 11 diagnostic statements and is partially balanced. Six items contain negative statements (2, 3, 5, 6, 9, 10), describing lack of satisfaction with social contacts (social loneliness). In contrast, the other five statements are positive (1, 4, 7, 8, 11) and they measure satisfaction from one’s interpersonal relations (emotional loneliness). A participant responds to what extent these statements describe their present situation and feelings. The answers are given on a five-point Likert scale, and they indicate: 1–definitely yes, 2–yes, 3–neither yes nor no, 4–no, 5–absolutely not. The general sense of loneliness index is calculated by re-encoding six negative items and summing all the test items. A higher total score for a respondent reflected a higher sense of loneliness. The overall scores range from 11 to 55 points. The scores were also analysed separately for two categories of loneliness described as social loneliness and emotional loneliness. The scale is highly reliable and homogeneous: the internal stability index (Cronbach alpha) is 0.89, inter-item correlation r = 0.42, and the homogeneous index H Loevinger–0.47 [33,34].

#### 2.3.2. Satisfaction with Life Scale (SWLS)

SWLS contains five statements to evaluate one’s satisfaction with life, expressed by how content one is with one’s achievements. A respondent answers to what extent they agree with each statement about their life to date on a 7-point scale: 1–I absolutely disagree, 2–I disagree, 3–I rather disagree, 4–I neither agree nor disagree, 5–I rather agree, 6–I agree, 7–I absolutely agree. The points are summed, and the total score denotes the level of satisfaction with one’s life. The scores lie within an interval from 5 to 35 points. The higher the score, the higher sense of satisfaction with one’s life. The SWLS has good psychometric properties, and the reliability index (Cronbach alpha) is 0.87 [24].

#### 2.3.3. Statistical Analysis

A statistical analysis of the data was performed with the Polish version of STATISTICA 13 (TIBCO, Palo Alto, CA, USA). The socio-demographic data were presented as the number of cases and percentage, and the group equipotency was verified with the chi-square test (χ^2^). The variables were described with the mean, standard deviation, median, minimum and maximum as well as the confidence interval for the mean ±95%. General satisfaction with life index was transformed into standardised units and interpreted according to the sten scale properties. Scores between 1 and 4 sten were regarded as low, whereas those from 7 to 10 sten were regarded as high, which corresponds to the area of ca. 33% of the lowest scores and the same percentage of the highest ones [24]. The scores of 5 and 6 sten were regarded as average. The diversity of the scores for a sense of loneliness and satisfaction with life among the students under study was evaluated by the analysis of variance ANOVA (F) of a comparison of many samples of independent groups. The Pearson correlation (r) was used to examine the significance of the strength of the correlation between the sense of loneliness and the satisfaction with life. A multiple regression analysis was performed in order to build a random variable estimation model from the independent variables. The interpretation of the correlation strength between the analysed variables was based on Guilford’s classification, taking, in sequence,: |r| = 0—no correlation, 0.0 < |r| ≤ 0.1—slight correlation, 0.1 < |r| ≤ 0.3—weak correlation, 0.3 < |r| ≤ 0.5—average correlation, 0.5 < |r| ≤ 0.7—high correlation, 0.7 < |r| ≤ 0.9—very high correlation, 0.9 < |r| < 1.0—nearly full correlation, |r| = 1—full correlation [35]. The level of significance of *p* < 0.5 was adopted in all the tests. The study meets the criteria of a cross-sectional study [36].

## 3. Results

### 3.1. Diversity of Scores for a Sense of Loneliness and Satisfaction with Life among Students in the Polish, Spanish and Slovak Samples

Using the Kolmogorov–Smirnov test, it was proven that the test values indicated a distribution of the variables close to a normal distribution in the studied samples. The data analysis revealed the overall intensity of loneliness among the nursing students in the study groups during the COVID-19 pandemic. No significant differences were found concerning a sense of loneliness among the students depending on the country of origin (F = 0.04; *p* < 0.96). The mean scores for the sense of loneliness on a general scale for the Polish (M = 25.07; SD = 9.40), Slovak (M = 25.19; SD = 8.24) and Spanish sample (M = 24.93; SD = 8.21) were at a similar level. No statistically significant diversity was found between the subgroups regarding social loneliness (F = 2.27; *p* < 0.10). However, significant differences were observed regarding the emotional loneliness level (F = 5.51; *p* < 0.04), which manifested itself as limited or no satisfying social contacts. The results of the difference significance test for these three subgroups are listed in Table 2.

In the subsequent step, detailed analyses with the post-hoc (NIR) test showed that the emotional loneliness index in the students in Spain was significantly lower than in those in Poland (*p* < 0.0001) and in Slovakia (*p* < 0.0001). However, no significant differences were found in the emotional loneliness level between students in Poland and Slovakia (Figure 1).

Significant differences between the subgroups can also be noted regarding the evaluation of a general index of satisfaction with life expressed as the degree of one’s contentment with one’s achievements and conditions (F = 30.19; *p* < 0.0001) (Table 1). The NIR test showed the level of general satisfaction with life index among the nursing students in the Polish group to be significantly lower than among the students in Spain (*p* < 0.0001) and Slovakia (*p* < 0.0001) (Figure 2).

The general satisfaction with life indexes for the Polish, Spanish and Slovak students was transformed into standardised units on the sten scale in statistical analyses. Statistically significant differences were shown to exist between the subgroups under comparison in the distribution of low, average and high scores (χ^2^ = 67.11; *p* < 0.0001). The structure of the scores showed that the percentage of students expressing a low satisfaction with life is higher in Poland (35.90%) than in Spain (21.43%) and Slovakia (12.94%). However, there were more students showing a high level of satisfaction with life in Spain (55.61%) and Slovakia (49.41%) than in Poland (27.95%) (Figure 3).

### 3.2. Strength of Correlations between a Sense of Loneliness and Satisfaction with Life among Students in the Polish, Spanish and Slovak Samples

Further analyses involved the calculation of Pearson’s linear correlation coefficients (r) between a sense of loneliness and satisfaction with life among nursing students in the Polish, Spanish and Slovak subgroups in connection with the consequences of the COVID-19 pandemic. All of the correlations under analysis proved to be statistically significant and negative, and their strength, according to J. Guilford’s classification, was high or average [35]. This means that the more strongly an individual suffers from loneliness, the lower their satisfaction with various aspects of life, which are important elements of health and vice versa. Highly negative correlations were found in the group of Polish students between a general sense of loneliness and satisfaction with life (r = −0.50; *p* < 0.001) and average negative correlations of social loneliness (r = −0.47; *p* < 0.001) and emotional loneliness (r = −0.47; *p* < 0.001). The sense of loneliness among the Spanish students had the strongest, statistically significant negative correlation with the sense of satisfaction with life at a high level, with respect to the general sense of loneliness (r = −0.53; *p* < 0.001) and social loneliness (r = −0.50; *p* < 0.001). A negative (but average) correlation was shown between emotional loneliness and satisfaction with life (r = −0.43; *p* < 0.001). The coefficients of correlation between a sense of loneliness and satisfaction with life among the Slovak students were negative, and the most significant (high) values were found for the general sense of loneliness (r = −0.51; *p* < 0.001) and (at an average level) for emotional loneliness (r = −0.46; *p* < 0.001), followed by social loneliness (r = −0.40; *p* < 0.001) (Figure 4).

### 3.3. Predictors of Social and Emotional Loneliness among Nursing Students in the Polish, Spanish and Slovak Samples

Further statistical analyses involved determining predictors of a sense of loneliness among the nursing students in each country. A sense of social loneliness and emotional loneliness were taken as a dependent variable in the process of constructing a multiple regression model. The pool of independent variables included socio-demographic variables, i.e., age, year of studies, place of residence during the COVID-19 pandemic, as well as satisfaction with life, reducing interpersonal contacts, time spent daily in front of a computer, a subjective health status assessment, number and regularity of meals and a restriction of physical activity. However, the statistical analyses showed that part of the independent variables did not have any impact on the regression model structure and, ultimately, the following were included: age, subjective health status assessment, reduction in social contacts, number of meals, a restriction of physical activity and satisfaction with life.

The regression analysis found three independent variables to be predictors of social loneliness among Polish students, explaining 26% of the score variation altogether. The regression coefficient was negative (ßeta = −0.40; R^2^ = 0.26), which is indicative of the negative correlation. A sense of satisfaction with life had the largest share (22%). The other two variables concerning the health status assessment and reduction in social contacts during the COVID-19 pandemic had a small share in predicting social loneliness among the Polish students (4%) (Table 3). Only one variable, i.e., a sense of satisfaction with life, was a predictor of social loneliness among the nursing students in Spain. Its predictive power explained 25% (ßeta = −0.49; R^2^ = 0.25) of the score variation (Table 3). The regression analysis among the Slovak students showed three independent variables, explaining 24% of the score variation, to be the determinants of social loneliness, but the sense of satisfaction with life had the largest share (16%). The regression coefficient was negative (ßeta = −0.40; R^2^ = 0.16), which is indicative of a negative correlation. The variable concerning the reduction in interpersonal contacts during the COVID-19 pandemic had a 5% share in the prediction of social loneliness, whilst this share for the frequency of meals was 3% (Table 4).

According to the results shown in Table 4, emotional loneliness in the group of Polish students was explained by two variables at a level of 24% of the dependent variable variation, and a sense of satisfaction with life was its significant predictor, which explained 22% of the results (ßeta = −0.42; R^2^ = 0.22). Another variable concerned a subjective health status assessment, and it explained 2% of the score variation, and it did not play a significant role in the prediction of emotional loneliness. The regression coefficient was negative in both cases, which was indicative of a negative correlation. The greatest contribution in the prediction of emotional loneliness among the Spanish students (21%) was made by a sense of satisfaction with life, which was also negative (ßeta = −0.40; R^2^ = 0.21). The variable which determined the intensity of the students’ physical activity during the COVID-19 pandemic explained 3% of the score variation. The situation in the Slovak group was similar—a sense of satisfaction with life also explained 21% of the dependent variable variation, and it was negative (ßeta = −0.47; R^2^ = 0.21). Another variable determining the age of the Slovak students explained 3% of the score variation in the prediction of emotional loneliness.

In conclusion, it can be claimed that satisfaction with life, which is an important element of health (and sometimes it is even equated with it), is the main predictor of a sense of loneliness in nursing students in Poland, Spain and Slovakia in the second year of the COVID-19 pandemic. All the regression models developed for this study were statistically significant (Table 3 and Table 4).

## 4. Discussion

A characteristic role in an assessment of a sense of loneliness among nursing students during the COVID-19 pandemic is played by external circumstances related to the consequences of the SARS-CoV-2 virus spread, which include reduction or blocking contacts, thereby eliminating so-called “reciprocal behaviours”. No diversity was observed in this study regarding a general sense of loneliness for the Polish, Spanish and Slovak samples, whereas differences were demonstrated in the level of emotional loneliness among the students. The lowest emotional loneliness indices were noted for the students in Spain. It was shown in a German study conducted by Diehl et al. that emotional loneliness was more common than social loneliness among the students and both variables were correlated positively with a sense of depression and anxiety [37].

Since the outbreak of the COVID-19 pandemic, students have been burdened with considerable changes in their everyday lives and online teaching. These stressors have made young people experience psychological issues [38]. The literature devoted to students’ mental health presents studies that deal with this subject matter from a broader perspective. It was shown in an Italian study that students experienced a high level of anxiety and stress, concentration and sleep disorders [38]. Other research findings may also raise concern as they confirm that health issues among students during the COVID-19 pandemic included depression, fear, anxiety, stress and sleep disorders [39,40,41,42]. There have been some reports by Ochnik et al., who examined differences in the mental health status between university students in nine countries during the first wave of the COVID-19 pandemic (Colombia, Czechia, Germany, Israel, Poland, Russia, Slovenia, Turkey, Ukraine). The researchers revealed differences in the occurrence of depression and anxiety between the study participants with respect to their country of residence. The Turkish students proved to be the most susceptible to experiencing depression and anxiety. The lowest depression indices were noted for the students in Czechia, and the lowest anxiety indices were found for the students in Germany. Significantly higher scores regarding the analysed variables were noted for the students in Poland than those in Czechia, Slovenia, Ukraine and Germany, and those in Russia regarding anxiety scores [43].

Nearly ⅓ of the students in Spain and ⅕ of the students in Poland claimed to spend ≥10 h daily in front of a computer in this study due to hybrid teaching. Over half of the respondents confirmed that their social contacts were considerably reduced during the COVID-19 pandemic. It was demonstrated in the regression analysis that the variable concerning social contact reduction during the pandemic had a 5% and 1% share in predicting social loneliness in the Slovak and Polish student groups, respectively. It should be noted that at the time of the study, students in Slovakia were only receiving distance learning due to the declared national emergency state. Thus, it can be concluded that the limitation of interpersonal contact, sometimes lasting many months during a pandemic, leads to a reduction in social networks and lack of social ties and, consequently, to loneliness subjectively perceived by students as social isolation.

Due to a reduction in direct contact with other people during the COVID-19 pandemic, a mobile phone became an intermediary in interpersonal contacts, and social media replaced real-world peer/social relations. Many studies have demonstrated that excessive focus on using a mobile phone can have a detrimental impact on psycho-social health. Sönmez et al. found a positive correlation among nursing students between addiction to smartphone use and loneliness [44]. A correlation between loneliness and Internet usage and between loneliness and the number of hours spent online among students in the Middle East (Kuwait, Saudi Arabia) was demonstrated by Alheneidi et al. those who claimed to feel lonelier often received information on the pandemic from social media [45]. It was demonstrated in a study with students in Bangladesh that loneliness and mental stress were positively correlated with problematic Internet use, whereas self-esteem was correlated negatively [46]. Another study conducted by Turan et al. did not find any significant correlation between students’ addiction to the Internet, loneliness and satisfaction with life. However, a significant correlation was observed between loneliness and satisfaction with life [47].

The current study revealed differences between participants in terms of satisfaction with life. There were significantly more students showing a high level of satisfaction with life in Spain and Slovakia than in Poland. However, this variable should be considered in the context of cultural differences. According to the authors of these findings, satisfaction with life is the main predictor of a sense of social and emotional loneliness among nursing students in Poland, Spain and Slovakia. In line with the hypothesis, it was confirmed that people with a high sense of life satisfaction experienced loneliness less frequently than those with a low level. Deutrom et al. reported similar results [27].

In the light of the research results obtained, it can also be hypothesised that students who are satisfied with their lives cope in a more adaptive way. This, in turn, strengthens their sense of quality of life and is probably related to their ability to cope with loneliness. It would be interesting to conduct further research to verify this hypothesis based on the achievements to date [48,49,50,51,52].

Şimşek et al. conducted a study in Turkey and showed that loneliness was a significant determinant next to cognitive distortions and satisfaction with life. The moderation analysis revealed some age-related differences between loneliness and satisfaction with life. The researchers observed that both loneliness and satisfaction with life increased with age. However, satisfaction with life among young people decreased as a sense of loneliness increased. It was concluded that although young people had more opportunities than older ones, their ability to cope with negative situations, such as loneliness, is underdeveloped to a great extent [48]. Clark et al. studied a Maltese population and demonstrated elevated loneliness indices among teenagers, which decreased slightly at the young adult age to increase from the age of 35. They proved that loneliness was also significantly correlated with a self-assessment of one’s physical health status, abilities to cope and subjective well-being [53]. Nursing students experience high levels of stress during the COVID-19 pandemic. This was confirmed, for example, in a study conducted among female nursing students in the Philippines. However, the students’ resistance, satisfaction with life and mental well-being ranged from moderate to high. Further, the increased pandemic-related stress reduced their satisfaction with life and caused their mental well-being to deteriorate [54].

Reviews of evidence of COVID-19 pandemic-related restrictions indicated that the perceived stress could have a negative impact on future professional choices among medical students, and the feeling of loneliness can have a mediating effect [55]. Clinical/practical education in the nursing study major during the COVID-19 pandemic requires large efforts from the organisers and academic teachers to adapt to all the restrictions related to the SARS-CoV-2 virus spread. Students improve their practical skills and develop their social competencies in direct contact with patients. Empathy, teamwork and lifelong education are specific elements of professionalism in medicine and are of particular importance in interactions with patients and in the well-being of medical personnel at the workplace. The findings of a study conducted by Berduzco-Torres et al. confirmed that loneliness has a detrimental impact on empathy development among nursing students [56].

This study demonstrated a decreased level of physical activity among nursing students. Only one in four students declared that their physical activity had not been reduced during the COVID-19 pandemic. The regression analysis showed that the variable which determined the intensity of the students’ physical activity of the Spanish students explained 3% of the score variation in emotional loneliness prediction. Xiang et al. analysed the physical activity of Chinese students and its correlates, such as anxiety and depression, and found that 52.3% of the study participants did not have sufficient physical activity. A high level of physical activity correlated significantly with decreased anxiety and depression indices. Moreover, specific types of physical exercise, such as stretching and workouts, were correlated negatively with both anxiety and depression [57].

The issue of loneliness among nurses at the workplace should also be emphasised. Loneliness at the workplace is related to restricted social interactions with leaders and decreased trust in leaders [58]. Moreover, loneliness and professional alienation lead to the deterioration of their professional performance [59]. Loneliness is a complex concept that includes both psychological and social aspects.

The latest research on challenges to mental health presented by the COVID-19 pandemic indicated the development of various support forms. The proposed interventional measures should be addressed to individuals at every age [53]. Academic Psychological Support Centres at universities are an interesting initiative. The proposed support includes emergency advisory interventions over the Internet and psychological interventions, i.e., psychoeducation, relaxation, creating day schedules, cultivating relations [38,60].

### Limitations and Implications Regarding Professional Practice

The considerations presented in this paper determined some limitations and implications for professional practice. First, they showed that the loneliness felt by young people is a challenge to their life and health. Seconded, there is a need to include some measures in the teaching activities aimed at recognising the needs and adaptation potential in students and providing support in difficult psychosocial situations. Third, it is important to care about subjective well-being, which reflects one’s health status. It could be valuable to enable students to make use of emergency psychological assistance or peer and family support. It is also necessary to take systemic/institutional prophylactic measures to prevent loneliness among young people during and after the COVID-19 pandemic.

This is the first international study conducted during the COVID-19 pandemic among nursing students in Poland, Spain and Slovakia. However, it has some limitations, including the group size, and it requires replication with larger study groups observed for longer periods. The results of this study could then be generalised to a larger population. Further limitations relate to the method and distribution of the questionnaires. The study was conducted in a mixed design, with students from Slovakia completing the survey questionnaires electronically, which may have influenced the results of the study. In addition, students were not excluded who had, among other things, family or financial problems not related to studying at the time of the survey.

## 5. Conclusions

Various levels of a sense of emotional loneliness and satisfaction with life were observed among nursing students in Poland, Spain and Slovakia. Emotional loneliness was felt to a lesser extent by students in Spain than by those in Poland, whereas no differences were found in the level of emotional loneliness between the students in Poland and Slovakia.

The overall level of satisfaction with life among students in Poland proved to be lower than in students in Spain and Slovakia.

Negative (average and high) correlations were found between a sense of loneliness and satisfaction with life among students in the Polish, Spanish and Slovak samples. Students with a stronger sense of loneliness also felt lower satisfaction with various aspects of life.

Satisfaction with life proved to be the main predictor of loneliness among nursing students in Poland, Spain and Slovakia in the second year of the COVID-19 pandemic.

It is important to take preventive and prophylactic actions concerning loneliness in students during and after the COVID-19 pandemic. Attempts to resume one’s previous lifestyle despite the COVID-19 pandemic can be a protective measure for maintaining one’s mental health.

## Figures and Tables

**Figure 1 ijerph-19-02929-f001:**
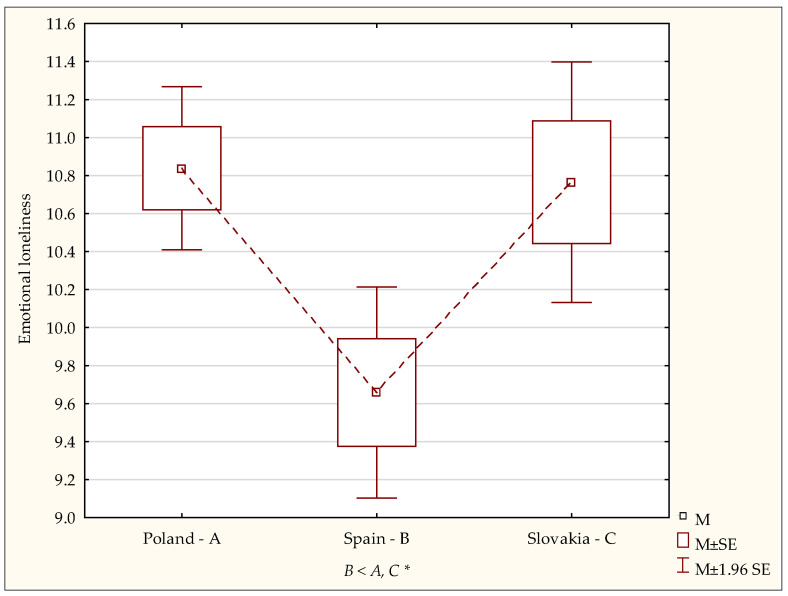
Emotional loneliness—diversity of scores. A,B,C—post-hoc analysis (NIR test). M—mean, SE—standard error. Statistically significant: * *p* < 0.05.

**Figure 2 ijerph-19-02929-f002:**
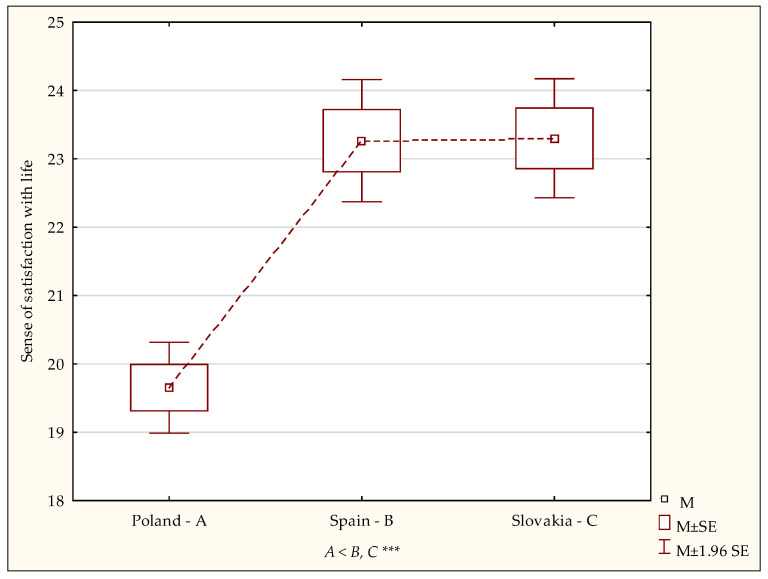
Sense of satisfaction with life—diversity of scores. A,B,C—post-hoc analysis (NIR test). M—mean, SE—standard error. Statistically significant: *** *p* < 0.001.

**Figure 3 ijerph-19-02929-f003:**
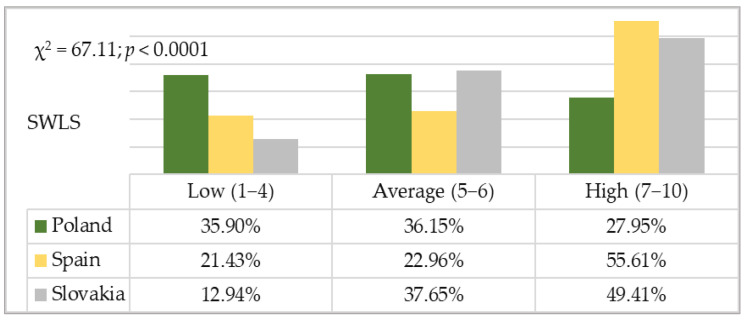
Structure scores for a sense of satisfaction with life on the sten scale in the Polish, Spanish and Slovak samples.

**Figure 4 ijerph-19-02929-f004:**
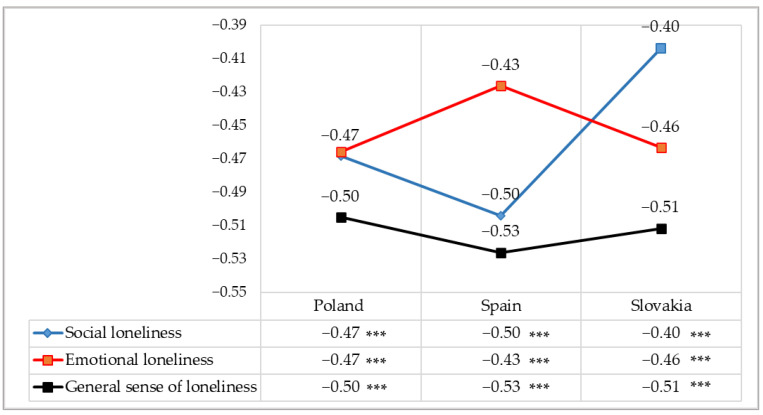
Nature and strength of the correlation between a sense of loneliness and satisfaction with life in the study participants–Pearson correlation coefficients (r). Statistically significant: *** *p* < 0.001.

**Table 1 ijerph-19-02929-t001:** Characteristics of participants.

Variables		Country of Residence N = 756		
		Poland—A *n* = 390	Spain—B *n* = 196	Slovakia—C *n* = 170
		*n* (%)	*n* (%)	*n* (%)
Gender	female	357 (91.54)	161 (82.14)	164 (96.47)
	male	33 (8.46)	35 (17.86)	6 (3.53)
Study year	first	140 (35.90)	28 (14.29)	55 (32.35)
	second	160 (41.03)	73 (37.24)	61 (35.88)
	third	90 (23.08)	95 (48.47)	64 (37.65)
Age (years)	≤20	128 (51.28)	63 (32.14)	71 (41.76)
	21–22	200 (44.18)	29 (14.80)	36 (20.59)
	≥23	62 (15.90)	126 (16.67)	126 (16.67)
Place and form of residence	with family/someone close	297 (76.15)	192 (97.96)	141 (82.94)
	on their own	93 (23.85)	4 (2.04)	29 (17.06)
Time of work on a computer (hours)	≤5	174 (44.62)	51 (26.02)	102 (60.00)
	6–9	135 (34.62)	84 (42.86)	55 (32.35)
	≥10	81 (20.77)	61 (31.12)	13 (7.65)
Number of meals	1–2	32 (8.21)	5 (2.55)	10 (5.88)
	3	174 (44.62)	47 (23.98)	45 (26.47)
	4	125 (15.13)	94 (47.96)	74 (43.53)
	≥5	59 (19.84)	50 (25.51)	41 (24.12)
Restriction of physical activity during the pandemic	no	89 (22.82)	69 (35.20)	43 (25.29)
	yes, to a small extent	66 (16.92)	51 (26.02)	54 (31.76)
	yes, to a medium extent	129 (33.08)	55 (28.06)	50 (29.41)
	yes, to a considerable extent	106 (27.18)	21 (10.71)	23 (13.53)
Subjective health status assessment during the pandemic	bad	9 (2.31)	4 (2.04)	2 (1.28)
	good/average	257 (65.90)	118 (60.20)	114 (67.06)
	very good	124 (31.79)	74 (37.76)	54 (31.76)
Restriction of social contacts during a pandemic	very high	82 (21.03)	98 (50.00)	36 (21.18)
	considerable	155 (39.74)	39 (19.90)	53 (31.18)
	medium/average	83 (21.28)	56 (28.57)	49 (28.82)
	to a small extent	70 (17.95)	3 (1.53)	32 (18.82)

Explanation: N–number of group members; *n*–number of subgroup members.

**Table 2 ijerph-19-02929-t002:** Sense of loneliness and satisfaction with life among the respondents in the study group—results of the difference significance test, taking into account the grouping variable—country of residence.

Variables		Country of Residence			ANOVA (F)	*p*
		Poland—A *n* = 390 (51.59%)	Spain—B *n* = 196 (25.93%)	Slovakia—C *n* = 170 (22.49%)		
		M, SD, Me, Minimum–Maximum, 95 CI	M, SD, Me, Minimum–Maximum, 95 CI	M, SD, Me, Minimum–Maximum 95 CI		
Sense of loneliness	General sense of loneliness	25.07, 9.40, 24, 11–49, 24.13, 26.00	24.93, 8.21, 24, 11–49, 23.77, 26.08	25.19, 8.24, 26, 11–50, 23.94, 26.44	0.04	0.96
	Social loneliness	14.23, 5.84, 13, 6–30, 13.65, 14.81	15.27, 5.21, 15, 6–28, 14.53, 16.00	14.42, 5.61, 14, 6–30, 13.57, 15.27	2.27	0.10
	Emotional loneliness	10.84, 4.32, 10, 5–25, 10.41, 11.27	9.66, 3.97, 9, 4–21, 9.10, 10.22	10.76, 4.21, 10, 5–21, 10.13, 11.40	5.51	0.04 * B < A,C
Sense of satisfaction with life		19.65, 6.69, 20, 5–35, 18.98, 20.32	23.27, 6.38, 24, 6–35, 22.37, 24.16	23.30, 5.79, 23, 5–35, 22.42, 24.18	30.19	0.0001 *** A < B,C

Statistically significant: * *p* < 0.05; *** *p* < 0.001; A,B,C—post-hoc analysis (NIR test). M—mean, SD—standard deviation, Me—median, 95% confidence interval (CI).

**Table 3 ijerph-19-02929-t003:** Summary of regression—a sense of social loneliness among nursing students in the Polish, Spanish and Slovak samples.

Group/ Country	Variables	R^2^	ßeta	ß	Error of ß	t	*p*
Poland	Constant value			26.66	1.96	13.61	0.0001 ***
	SWLS	0.22	−0.40	−0.35	0.04	−8.69	0.0001 ***
	Subjective health status assessment	0.25	−0.14	−1.12	0.37	−3.04	0.003 ***
	Reduction in social contacts during the pandemic	0.26	−0.11	−0.64	0.27	−2.40	0.02 *
	R = 0.52; R^2^ = 0.27; corrected R^2^ = 0.26						
Spain	Constant value			23.70	1.52	15.59	0.0001 ***
	SWLS	0.25	−0.49	−0.40	0.05	−7.78	0.0001 ***
	R = 0.51; R^2^ = 0.26; corrected R^2^ = 0.25						
Slovakia	Constant value			28.61	3.04	9.42	0.0001 ***
	SWLS	0.16	−0.40	−0.36	0.07	−5.31	0.0001 ***
	Reduction in social contacts during the pandemic	0.21	−0.26	−1.02	0.39	−2.63	0.01 **
	Number of meals	0.24	−0.18	−0.84	0.46	−1.82	0.07
	R = 0.50; R^2^ = 0.25; corrected R^2^ = 0.24						

Statistically significant: * *p* < 0.05; ** *p* < 0.01; *** *p* < 0.001. R—correlation coefficient, R^2^—multiple determination coefficient, ßeta–standardised regression coefficient, ß—non-standardised regression coefficient, Error ßeta–non-standardised regression coefficient error, t—*t*-test value.

**Table 4 ijerph-19-02929-t004:** Summary of regression—a sense of emotional loneliness among nursing students in the Polish, Spanish and Slovak samples.

Group/ Country	Variables	R^2^	ßeta	ß	Error of ß	t	*p*
Poland	Constant value			19.90	1.39	14.28	0.0001 ***
	SWLS	0.22	−0.42	−0.27	0.03	−9.01	0.0001 ***
	Subjective health status assessment	0.24	−0.13	−0.79	0.27	−2.90	0.004 ***
	R = 0.49; R^2^ = 0.24; corrected R^2^ = 0.24						
Spain	Constant value			14.10	1.20	11.74	0.0001 ***
	SWLS	0.21	−0.40	−0.25	0.04	−6.08	0.0001 ***
	Physical activity during the pandemic	0.24	0.16	0.61	0.25	2.41	0.01 **
	R = 0.45; R^2^ = 0.20; corrected R^2^ = 0.24						
Slovakia	Constant value			8.67	3.66	2.37	0.02 *
	SWLS	0.21	−0.47	−0.34	0.05	−6.77	0.0001 ***
	Age	0.24	0.16	0.32	0.14	2.29	0.02 *
	R = 0.51; R^2^ = 0.26; corrected R^2^ = 0.24						

Statistically significant: * *p* < 0.05; ** *p* < 0.01; *** *p* < 0.001. R—correlation coefficient, R^2^—multiple determination coefficient, ßeta—standardised regression coefficient, ßeta—non-standardised regression coefficient, Error ßeta—non-standardised regression coefficient error, t—*t*-test value.

## Data Availability

The data presented in this study are available on request from the first author.

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
