# Peer review of "Loneliness and Satisfaction with Life among Nursing Students in Poland, Spain and Slovakia during the COVID-19 Pandemic"

_ijerph, 2022, doi:10.3390/ijerph19052929_

Round 1
Reviewer 1 Report
This paper is interesting and timely important. However, the follwoing minor comments should be addressed.
- Introduction : The authors must explain why this paper is important. In particular, theoretical justifications are essential.
- If possible, two or more research hypotheses are critical for a better understanding of the work.
Author Response
Dear Reviewers, Thank you very much for a thorough editorial assessment of my manuscript, positive opinions, as well as the reviewers’ remarks. I used them as an important guide to improving the quality of my paper. The implemented corrections were done strictly according to their comments. All changes made in the text are marked in yellow. I have enclosed the re-edited manuscript and cover letter as responses to Reviewers, detailing how I followed their suggestions. Thank you very much for your kind consideration of my paper. Yours sincerely, Ewa Kupcewicz, PhD REV. 1 1. This paper is interesting and timely important. However, the follwoing minor comments should be addressed. Thank you. 2. Introduction : The authors must explain why this paper is important. In particular, theoretical justifications are essential. In the Introduction section, the authors tried to explain why the undertaken research was important. The modified text was highlighted in yellow. 3. If possible, two or more research hypotheses are critical for a better understanding of the work. As suggested by the Reviewer, a research hypothesis was introduced.
Reviewer 2 Report
The authors completed a study to determine a correlation between loneliness and a satisfaction with life among nursing students in Poland, Spain and Slovakia and to seek predictors of social and emotional loneliness among the students as a consequence of the spread of the SARS-CoV-2 virus. Even though the idea for this study is good and seems to fulfil a gap in current knowledge, the way it is presented does not help the reader to understand why this study is needed. Specifically, it is not clear in the introduction what is the knowledge gap that this study fills. The introduction includes a lot of information without linked it with the aim. I see that COVID-19 restrictions may be an opportunity to explore these factors, the question is why to do this. Instead, in the discussion it is clearer why this study is needed. However, the discussion is a descriptive text of previous evidence. The only explanation of the outcomes of the current study is the “cultural differences” between students who live in different countries. The reader would expect, a comprehensive analysis of the measures taken in each country during the pandemic that may explain these differences. Consequently, the reader would also expect to see, what are the differences in the factors that were analyzed during the pandemic with pre-pandemic periods, in this population. In this light, I think that the paper suffers from clarity. Please also see some minor comments below.
The <30 age criterion (line 107) is not actually mentioned in the introduction and why is this needed. I think that the information is there, however, the way that is presented does not actually help the reader.
Why the acronym “Approx.” is used instead of approximately (lines 115, 139).
Regarding the characteristics of the participants, a table would be more useful, rather than a long description in the text
Line 153: “Physical exercise”. Do you mean physical activity levels? How was this determined?
I see from the statistical analysis a parametric analysis approach. Did you determine normal distribution analysis, or you made assumptions. Either way, please elaborate
In the section “Limitations and implications regarding professional practice”, I only found implications, not limitations. This is important, given that the study has limitations.
Author Response
Dear Reviewers, Thank you very much for a thorough editorial assessment of my manuscript, positive opinions, as well as the reviewers’ remarks. I used them as an important guide to improving the quality of my paper. The implemented corrections were done strictly according to their comments. All changes made in the text are marked in yellow. I have enclosed the re-edited manuscript and cover letter as responses to Reviewers, detailing how I followed their suggestions. Thank you very much for your kind consideration of my paper. Yours sincerely, Ewa Kupcewicz, PhD

Reviewer 3 Report
This study aims to determine the correlation between loneliness and satisfaction with life among nursing students in Poland, Spain and Slovakia and to seek predictors of social and emotional loneliness among the students. It is an interesting subject, in particular with the participation of students across 3 countries. Some revisions regard:
In sample description (section 2.2) I suggest the creation of a Table to indicate major participants' characteristics (e.g., country, age, year of study, time spent on computer, living alone).
The headings of 3.1, 3.2 and 3.3 could be slightly modified: e.g., use the word "sample/participants" instead of "studies", since this is ONE STUDY including participants from 3 countries.
(Detailed) findings of earlier research are suggested to be discussed in literature review (or in introductory section). For example, on page 10, about nine lines are devoted to another study [31]; this should be presented early on, while in "Discussion" section the authors should discuss/interpret possible (dis)agreement with earlier research.
On page 7, line 253, besides "Guilford’s classification", add the relevant reference.
Since the variable 'context/country' is important, the authors could also mention whether the findings of this study could be generalizable; if yes, this has implications for future research in other countries. Future research studies could also administer the same research instrument(s) to detect similarities and differences among nursing students of different countries.
Author Response

(The authors gave the same response as above.)

Round 2
Reviewer 2 Report
The authors did not respond with a point by point letter, which makes it difficult to test the responses on my comments. From the modifications that I found in the revised paper I have the following comments:
- The explanations of the outcomes of the study given in the discussion, seems reasonable, however, there are not supported by references. Is there any other published studies in the area, not necessarily on the nurse population?
- Similarly, the additional information given in the introduction, regarding the need for the study to be conducted, are not supported by references.
- I am still confused with the "physical exercise" term. There are several parts of the paper refers to "physical activity" and some other to "physical exercise".
Author Response
The authors did not respond with a point by point letter, which makes it difficult to test the responses on my comments. From the modifications that I found in the revised paper I have the following comments:
Dear Reviewer, Thank you very much for the thorough editorial assessment of our manuscript, positive opinions, as well as reviewers’ remarks. Our corrections were implemented strictly according to the reviewer’s comments. All changes made in the text are marked in yellow. We are enclosing the re-edited manuscript and cover letter with responses to the reviewers, detailing how we followed their suggestions.
1. The explanations of the outcomes of the study given in the discussion, seems reasonable, however, there are not supported by references. Is there any other published studies in the area, not necessarily on the nurse population?
-1. Significant directions of considerations indicated by the reviewer in the Discussion section have been supplemented with a few literature items.
2. Similarly, the additional information given in the introduction, regarding the need for the study to be conducted, are not supported by references.
-2. Significant directions of considerations indicated by the reviewer in the Introduction section have been supplemented with a few literature items.
3. I am still confused with the "physical exercise" term. There are several parts of the paper refers to "physical activity" and some other to "physical exercise".
-3. The term should be „restrictions of physical activity during the pandemic”. This term has been corrected throughout the article.